# Natural Reweighted Wake-Sleep

**Csongor Várady**
Max Planck Institute for
Mathematics in the Sciences
Leipzig, Germany
varady@mis.mpg.de

**Riccardo Volpi**
Romanian Institute of
Science and Technology
Cluj-Napoca, Romania
volpi@rist.ro

**Luigi Malagò**
Romanian Institute of
Science and Technology
Cluj-Napoca, Romania
malago@rist.ro

**Nihat Ay**
Max Planck Institute for
Mathematics in the Sciences
Leipzig, Germany
nihat.ay@mis.mpg.de

## Abstract

Natural gradient has been successfully employed in a wide range of optimization problems. However, for the training of neural networks the resulting increase in computational complexity sets a limitation to its practical application. Helmholtz Machines are a particular type of generative models, composed of two Sigmoid Belief Networks, commonly trained using the Wake-Sleep algorithm. The locality of the connections in this type of networks induces sparsity and a particular structure for the Fisher information matrix that can be exploited for the evaluation of its inverse, allowing the efficient computation of the natural gradient also for large networks. We introduce a novel algorithm called Natural Reweighted Wake-Sleep, a geometric adaptation of Reweighted Wake-Sleep, based on the computation of the natural gradient. We present an experimental analysis of the algorithm in terms of speed of convergence and the value of the log-likelihood, both with respect to number of iterations and training time, demonstrating improvements over non-geometric baselines.

## 1   Introduction

Deep generative models have been successfully employed in unsupervised learning to model complex and high dimensional distributions thanks to their ability to extract higher-order representations of the data and thus generalize better [16, 8]. An approach which proved to be successful and thus common to several models is based on the use of two separate networks: the recognition network, i.e., the encoder, which provides a compressed latent representation for the input data, and the generative network, i.e., the decoder, able to reconstruct the observation in output. Helmholtz Machines (HM) [11] consist of a recognition and a generative network both modelled as a Sigmoid Belief Network (SBN) [22], and differently from standard VAEs [19, 23], are characterized by discrete hidden variables.

The training of stochastic networks is a challenging task in deep learning [12], in particular for generative models based on recognition and generative networks. A solution to this problem consists in the introduction of a family of tractable approximate posterior distributions parameterized by the encoder network. However, in the presence of discrete hidden variables, as for HMs, this approach cannot be directly employed, and thus standard training procedures relies on the well-known Wake-Sleep [15] algorithm, in which two optimization steps for the parameters of the recognition and generative networks are alternated. The Wake-Sleep algorithm, as well as more

34th Conference on Neural Information Processing Systems (NeurIPS 2020), Vancouver, Canada.

recent advances [10, 9, 24, 14], relies on the conditional independence assumption between the hidden variables of each layer, as a consequence of the directed graphical structure of a SBN. This leads to a computationally efficient formula for the weights update which does not require the gradients to be back-propagated through the full network. Besides the choice of the specific loss function to be optimized, depending on the nature of the generative model, in the literature several approaches to speed up the convergence during training have been proposed, through the definition of different training algorithms. One line of research, initiated by Amari and co-workers [3, 2], takes advantage of a geometric framework based on notions of Information Geometry [5] which allows the definition of the natural gradient, i.e., the Riemannian gradient of a function computed with respect to the Fisher-Rao metric. In general the computation of the natural gradient requires the inversion of an estimation of the Fisher information matrix, whose dimension depends on the number of weights, and for this reason it cannot be directly applied for the training of large neural network due to its computation costs.

Motivated by preliminary results from [6], we observe that the Fisher information matrix associated to a HM takes a block diagonal structure, where the block sizes depend linearly on the size of individual hidden layers. Starting from this observation, we propose a geometric version of the Reweighted Wake-Sleep algorithm for the training of HMs, where the gradient is replaced by the corresponding natural gradient.

## 2 The Natural Reweighted Wake-Sleep Algorithm

A Helmholtz Machine with $L$ layers is composed by two Sigmoid Belief Networks, a generative network $p$, parameterized by $\theta$ (a set of weights and biases $W^i$ for each layer $i$, with $i = 0, .., L$), and a recognition (or inference) network $q$, parameterized by $\phi$ (a set of weights and biases $V^i$ for each layer $i$, with $i = 0, .., L - 1$).

Following [10], the training of Helmholtz Machines can be recast in terms of a variational objective [13, 19, 23]. This is analogous to the learning in a Variational Autoencoder which requires the maximization of a lower bound for the likelihood. In the Reweighthed Wake-Sleep (RWS) [10, 20] algorithm, the derivative of the log-likelihood of $x$ are approximated with a Monte Carlo sampling in which the samples are reweighted [10, 20] with the importance weigths $\tilde{\omega}_k$, i.e.,

$$\frac{\partial \ln p(x)}{\partial \theta} = \frac{1}{p(x)} \mathbb{E}_{h \sim q(h|x)} \left[ \frac{p(x,h)}{q(h|x)} \frac{\partial \ln p(x,h)}{\partial \theta} \right] \simeq \sum_{k=1}^{K} \tilde{\omega}_k \frac{\partial \ln p(x, h^{(k)})}{\partial \theta} \quad \text{with} \ h^{(k)} \sim q(h|x) \ .$$

(1)

$$\tilde{\omega}_k = \frac{\omega_k}{\sum_{k'} \omega_{k'}} \ , \ \text{with} \ \omega_k = \frac{p(x, h^{(k)})}{q(h^{(k)}|x)} \ .$$

(2)

The RWS algorithm also introduces another training step called the q-wake update, additionally to the wake and sleep phases to outperform the regular WS. Our method builds on the RWS to calculate the gradients for each step.

Information Geometry [1, 5, 4, 7] studies the geometry of statistical models using the language of Riemannian and affine geometry. In this framework, the steepest direction of a function is given by the natural gradient update step, computed by

$$\theta_{t+1} = \theta_t - \eta F(\theta)^{-1} \nabla \mathcal{L}(\theta) \ ,$$

(3)

where $\mathcal{L}$ is the loss function and $\nabla \mathcal{L}$ its vector of partial derivatives. $F(\theta)$ is the Fisher information matrix associated to the Riemannian Fisher-Rao metric on the manifold of probability distributions parameterized by $\theta$, i.e., the weights of the network. Finally, $\eta > 0$ is the learning rate.

The Fisher information matrix $F$, needed for the evaluation of the natural gradient of a given function, has a structure which strongly depends on the nature of the statistical model. For both Sigmoid Belief Networks (SBNs) constituting a Helmholtz Machine, the Fisher information matrix can be computed as follows. Let $i$ denote a layer, and $j$ one of its hidden nodes, we denote with $W_j^i$ the $j$-th column of the matrix $W^i$. The blocks associated to the $i$-th layer and its $j$-th hidden unit for both the distributions $p$ and $q$ associated to the generative and the recognition networks, are denoted with $F_{p,j}^i$ and $F_{q,j}^i$, respectively. These matrices can be estimated using Monte Carlo methods, based on the

samples in the batch.

$$F_{p,j}^i = \mathbb{E}_{p(x,h)}\left[\sigma'\left(W_j^i h^{i+1}\right) h^{i+1} h^{i+1\,\mathrm{T}}\right] \approx \frac{1}{n}\sum_k \tilde{\omega}_k \sigma'\left(W_j^i h^{i+1\,(k)}\right) h^{i+1\,(k)} h^{i+1\,(k)\,\mathrm{T}} \ , \quad (4)$$

$$F_{q,j}^i = \mathbb{E}_{q(x,h)}\left[\sigma'\left(V_j^i h^{i-1}\right) h^{i-1} h^{i-1\,\mathrm{T}}\right] \approx \frac{1}{n}\sum_k \sigma'\left(V_j^i h^{i-1\,(k)}\right) h^{i-1\,(k)} h^{i-1\,(k)\,\mathrm{T}} \ . \quad (5)$$

The Fisher matrix is block-diagonal, as it can be seen in Fig. 1 and all other blocks besides $F_{p,j}^i$ and $F_{q,j}^i$ are zero by definition. Such block structure is very convenient and represents the main argument for the efficiency of the algorithm. Notice that the estimations in Eqs. (4) and (5) are not to be confused with the approximations typically introduced for the simplification of the Fisher matrix needed to make it computationally tractable in feed-forward neural networks. Each block of the Fisher matrix is estimated by Monte Carlo sampling based on $n$ samples. To obtain a lower variance estimator for the expected value in Eq. (4) we use samples $h^{(k)} \sim q(h|x)\,p_{\mathcal{D}}(x)$ from the approximate distribution, and reweight them by the importance sampling weights $\tilde{\omega}_k$ as in Eq. (2).

Since the Fisher matrices in Eqs. (4) and (5) only depend on the statistical models associated to the joint distributions $p(x,h)$ and $q(x,h)$, thus the model of the Helmholtz Machine remains unchanged, the same Fisher matrix can be used for different training algorithms (WS, RWS, etc.).

We introduced a matrix representation, where the $H^i$ matrices are obtained by concatenating for each sample the $h^i$ vector as a row vector, while the diagonal matrices $Q_p^i$ and $Q_q^i$ depends on the evaluation of the activation function. The matrices $H^{\mathrm{T}}QH$ associated with the estimation of the Fisher blocks may be singular depending on the number of samples in the batch used in the estimation and the size $l_i$ of each layer $i$. During training $n$ is the size of the mini-batch $b$ multiplied by the number of samples $s$ from the network (respectively $p$ or $q$, depending on the Fisher matrix under consideration). Notice that during training typically $n < l_i$, thus to guarantee the invertibility we add a damping factor $\alpha > 0$ multiplying the identity matrix in addition to $H^{\mathrm{T}}QH$. Now we can use the Shermann-Morisson formula to calculate the inverse of a rank-$k$ update matrix by

$$\tilde{F}^{-1} = \left(\frac{\alpha \mathbb{1}_l + H^T Q H}{1 + \alpha}\right)^{-1} = \frac{1+\alpha}{\alpha}\left(\mathbb{1}_l - H^T(\alpha Q^{-1} + H H^T)^{-1} H\right) \ , \quad (6)$$

so that $\tilde{F}^{-1} \to \mathbb{1}_l$ for $\alpha \to \infty$ and $\tilde{F}^{-1} \to F^{-1}$ for $\alpha \to 0$, if $F$ is full rank. By keeping in memory the rank-$k$ representation of the matrices, the overall complexity is $\mathcal{O}(l_0 l_1 n^2)$ where $l_0$ and $l_1$ are the bottom two layers of the Helmholtz Machine, which are usually the largest.

Assuming the locality of the gradient descent learning step, we make the approximation of slowly changing metric during few training steps. Under these assumptions we can reuse the Fisher matrix for a certain amount of steps $K$, without recalculating it. We call this technique, K-step update.

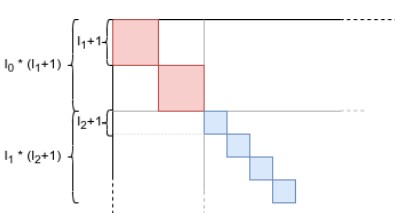

Figure 1: Block structure of the Fisher information matrix for Deep Sigmoid Belief Networks.

| alg | s | lr | LL | T/E |
|---|---|---|---|---|
| WS | 10 | 0.002 | -89.84 | 37s |
| RWS | 10 | 0.002 | -87.35 | 50s |
| NRWS | 10 | 0.001 | -84.91 | 250s |
| VAE [19] | - | - | ≈ -89.5 | - |
| RWS [10] | 10-100 | 0.001-0.0003 | ≈ -86.0 | - |
| BiD [9] | 10-100 | 0.001-0.0003 | ≈ -85.0 | - |

Table 1: Importance Sampling estimation of the log-likelihood (**LL**) with 10,000 samples for different algorithms after training till convergence with SGD. The damping factor used is 0.2. **T/E** - average time per epoch; **s** - samples in training.

## 3   Experiments

We use the binarized version of the MNIST database of handwritten digits [21]. The model architecture is a binary Helmholtz Machine with layers of sizes 300, 200, 100, 75, 50, 35, 30, 25, 20,

15, 10, 10, as in Bornschein et al. [9]. The training is performed without data augmentation, with binary variables in $\{-1, 1\}$. We used a mini-batch size of 32 for all experiments and no regularizers nor decaying learning-rate. All experiments were run with CUDA optimized Tensorflow on Nvidia GTX1080 Ti GPUs.

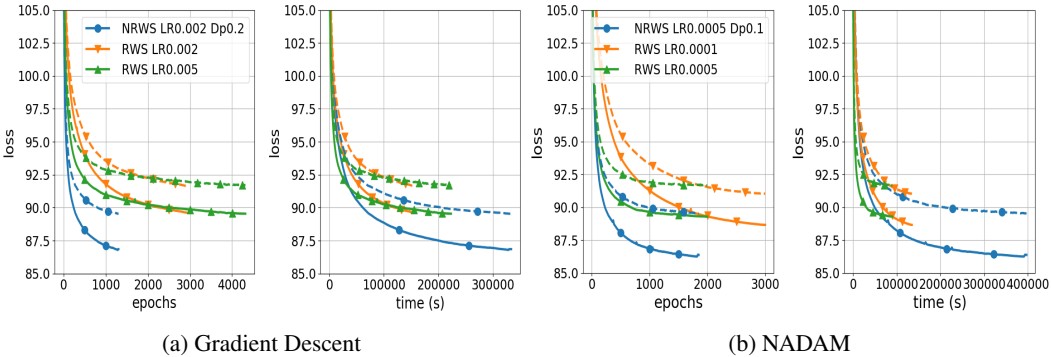

(a) Gradient Descent                      (b) NADAM

Figure 2: Training curves with (a) Gradient Descent and (b) Nesterov Adaptive Momentum (NADAM), continuous lines represent the quantities on the train set, and dashed lines the ones on validation; Left: Loss of algorithms in epochs; Right: Loss of algorithms in wall-clock time (s).

In Fig. 2a we present loss curves during training, for training and validation. The advantage of the NRWS in this case comes in the form of convergence to a better minimum. NRWS converges faster than the their non-geometric counterparts in epochs. Even increasing the learning rate of RWS it is not possible to achieve the same convergence speed (left panel of Fig. 2a). Even in time (right panel), NRWS is fast enough to compete with vanilla RWS. While NRWS outperformes the other methods, a possible drawback is that the in real-time it takes roughly 5 times as long to reach the same amount of epochs as its non-natural counterpart, still reaching a better optimum. Reducing the number of samples for the importance weighting results in faster convergence but it could also impact the performances. We compare our SGDs implementations of WS, RWS and NRWS with state-of-the-art [10, 9] in Table 1. Implementations from the literature also take advantage of accelerated gradient methods (ADAM[18]), learning-rate decay (from $10^{-3}$ to $3 \times 10^{-4}$), $L1$ and $L2$ regularizers and an increased number of samples towards the end of the training (from 10 to 100), in order to achieve better results. While further hyperparameter tuning could be successfully employed to improve our reported results, as well as variable learning rates, regularizers and number of samples [9], this is out of the scope of the present paper. Even with a simple training procedure (fixed learning rate, no regularization and fixed number of samples 10) we notice how the IS Likelihood on 10k samples is comparable and even better than RWS and BiHM as reported from the literature [10, 9].

Additionally, we tested the algorithms using the NADAM (Nesterov-Adam) optimizer [17, 25, 18]. In Fig. 2 we see that the NRWS also benefits from the accelerated gradient method outperforming in epochs as well as in real-world time the RWS. Similarly to SGD, the speed up of RWS by using a larger learning rate, does not help it catch up with NRWS. However, the adaptive steps and the accumulated momentum of NADAM are implicitly assuming an Euclidean metric in the tangent space, which is not the geometrically correct approach. This motivates the exploration of adaptive riemannian gradient methods for the NRWS algorithm, as a future work.

## 4 Conclusions

We showed how the graphical structure of Helmholtz Machines allows for the efficient computation of the Fisher matrix and thus the natural gradient during training, by exploiting the locality of the connection matrix. We introduced the Natural Reweighted Wake-Sleep (NRWS) algorithm and we demonstrated an improvement of the convergence during training for standard gradient descent over the state-of-the-art baselines WS and RWS. NRWS was not only faster to converge, but the obtained optimum resulted in better values for the IS likelihood estimation compared to the values reported with RWS and with BiHM. These results hold with a fixed number of samples and learning rate, without taking advantage of decaying learning rate and increasing number of samples during training as its literature counterparts. When using the NADAM optimizer, NRWS maintain the speed

advantage and the convergence to a better optimum, compared with its non-geometric counterpart. This encourages the exploration of adaptive gradient methods for the Natural Reweighted Wake-Sleep algorithm in which the Fisher-Rao matrix is explicit considered for the momentum accumulation and the adaptive step. Moreover, we plan to further study ways to obtain a better estimation of the Fisher matrix while training. As a final remark, we highlight that since the computation of the Fisher matrix is only dependent on the underlying statistical model, also other algorithms for the training of HMs could benefit from the use of the natural gradient, such as Bidirectional Helmholtz Machines.

## Acknowledgments and Disclosure of Funding

This work was supported by the DeepRiemann project, co-funded by the European Regional Development Fund and the Romanian Government through the Competitiveness Operational Program 2014-2020, Action 1.1.4, project ID P_37_714, contract no. 136/27.09.2016.

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
