# OpenReview forum: "Natural Reweighted Wake-Sleep"
_NeurIPS.cc/2020/Workshop/DL-IG — NeurIPSW 2020: DL-IG Poster_

### Official Review · AnonReviewer1 · 2020-10-26
**Review of "Natural Reweighted Wake-Sleep"**

**Rating:** 7
**Confidence:** 4

**Review:**

This paper proposes to speed up the training for Helmholtz machines by exploiting structure in the model to efficiently estimate the Fisher matrix for use with natural gradient descent. Helmholtz machines are networks with stochastic discrete hidden variables and are trained by estimating the objective through sampling and, in this paper, may include a sample re-weighting based on the mismatch between the encoder and decoder. The paper raises an interesting observation that the Fisher matrix for this model is block structured and can also be estimated efficiently through sampling. The experiments on MNIST suggest that using the Fisher matrix for natural gradient descent can speed up training.

This seems like an interesting exploration of how an alternative architecture and training approach could be made more competitive. Even with the small blocks in the Fisher matrix, I worry that the estimation could be quite noisy, and, as the authors point out, it does come with a significant computational cost. I'm wondering if the procedure could be made faster and more stable with a more aggressive approximation of F. For instance, could each block be considered as a diagonal plus rank one correction? Then, instead of using something like Tikhonov regularization as in Eq. 6, you might be able to use Sherman Morrison to directly invert each block (inverse of diagonal plus rank one is also diagonal plus rank one). One other comment is about the variance of the weights in the reweighted wake-sleep (and in Eq. 4 & 5). In early epochs of training, I would expect high variance for weights, and therefore also high variance for the Fisher matrix. Perhaps the smoothing could be sensitive to this - so that early iterations are more smoothed.

---

### Official Review · AnonReviewer2 · 2020-11-03
**Review of "Natural Reweighted Wake-Sleep"**

**Rating:** 8
**Confidence:** 4

**Review:**

The paper utilizes specific sparsity structure of the so-called Helmholtz machines to efficiently evaluate the natural gradients resulting into more efficient training of the machines. The paper is rigorous and computational experiments capture the essence of the algorithm. I recommend accepting the paper for the workshop.

---

### Author Response · Authors · 2020-12-09
**The link to the presentation on youtube**

The link to the video presentation of the poster:
https://youtu.be/yYC7w6E7z8E
On need I can also send the slides

---

### Decision · Program_Chairs · 2020-11-07

Accept (Poster)